# Diagnostic Yield and Economic Implications of Whole-Exome Sequencing for ASD Diagnosis in Israel

**DOI:** 10.3390/genes13010036

**Published:** 2021-12-23

**Authors:** Rotem Tal-Ben Ishay, Apurba Shil, Shirley Solomon, Noa Sadigurschi, Hadeel Abu-Kaf, Gal Meiri, Hagit Flusser, Analya Michaelovski, Ilan Dinstein, Hava Golan, Nadav Davidovitch, Idan Menashe

**Affiliations:** 1Department of Public Health, Faculty of Health Sciences, Ben-Gurion University of the Negev, Beer-Sheva 84100, Israel; rotem1@post.bgu.ac.il (R.T.-B.I.); apurba@post.bgu.ac.il (A.S.); 2Azrieli National Centre for Autism and Neurodevelopment Research, Ben-Gurion University of the Negev, Beer-Sheva 84100, Israel; ssolomon@post.bgu.ac.il (S.S.); nbabkoff@gmail.com (N.S.); hadeel.ak@gmail.com (H.A.-K.); galme@clalit.org.il (G.M.); hflusser@gmail.com (H.F.); dranalyami@clalit.org.il (A.M.); dinshi@bgu.ac.il (I.D.); havag@bgu.ac.il (H.G.); 3Zlotowski Center for Neuroscience, Ben-Gurion University of the Negev, Beer-Sheva 84100, Israel; 4Department of Physiology and Cell Biology, Faculty of Health Sciences, Ben-Gurion University of the Negev, Beer-Sheva 84100, Israel; 5Preschool Psychiatric Unit, Soroka University Medical Center, Beer-Sheva 84100, Israel; 6Child Development Center, Soroka University Medical Center, Beer-Sheva 84100, Israel; 7Psychology Department, Ben-Gurion University of the Negev, Beer-Sheva 84100, Israel; 8Department of Health Systems Management, Ben-Gurion University of the Negev, Beer-Sheva 84100, Israel; nadavd@bgu.ac.il

**Keywords:** autism spectrum disorder, genetics, whole-exome sequencing, diagnostic yield, cost-effectiveness analysis

## Abstract

Whole-exome sequencing (WES) is an effective approach to identify the susceptibility of genetic variants of autism spectrum disorder (ASD). The Israel Ministry of Health supports WES as an adjunct tool for ASD diagnosis, despite its unclear diagnostic yield and cost effectiveness. To address this knowledge gap, we applied WES to a population-based sample of 182 Bedouin and Jewish children with ASD from southern Israel, and assessed its yield in a gene panel of 205 genes robustly associated with ASD. We then compared the incremental cost-effectiveness ratios (ICERs) for an ASD diagnosis by WES, chromosomal microarray analysis (CMA), and CMA + WES. Overall, 32 ASD candidate variants were detected in 28 children, corresponding to an overall WES diagnostic yield of 15.4%. Interestingly, the diagnostic yield was significantly higher for the Bedouin children than for the Jewish children, i.e., 27.6% vs. 11.1% (*p* = 0.036). The most cost-effective means for genetic testing was the CMA alone, followed closely by the CMA + WES strategy (ICER = USD 117 and USD 124.8 per child). Yet, WES alone could become more cost effective than the other two approaches if there was to be a 25% increase in its yield or a 50% decrease in its cost. These findings suggest that WES should be recommended to facilitate ASD diagnosis in Israel, especially for highly consanguineous populations, such as the Bedouin.

## 1. Introduction

Autism spectrum disorder (ASD) is a collection of heterogeneous neurodevelopmental disorders that affects nearly 2% of children worldwide [1]. While the specific etiologies of ASD are still largely unknown, it is currently believed that ASD has a significant genetic susceptibility component, as is evident from multiple twin, familial and population studies [2,3,4,5,6,7]. Indeed, the genetic studies of ASD over the past two decades have revealed hundreds of rare genetic variants associated with ASD susceptibility [8], most commonly, the rare copy number variants (CNVs) [9,10,11,12,13] and single nucleotide variants (SNVs) [14,15,16]. To date, these variants together can explain up to 35% of ASD cases [17,18,19,20], and can be used for genetic screening of ASD [21]. In this realm, whole-exome sequencing (WES), which captures 1–2% of protein-coding sequences in the genome, is the genomic technique most relevant for clinical use [22,23].

In Israel, an ASD diagnosis is based on behavioral and cognitive assessments and is determined by a psychiatrist and a developmental pediatrician, or a child neurologist together with a clinical psychologist [24]. In addition, each child diagnosed with ASD is eligible for a government-funded chromosomal microarray (CMA) test [25]. Recently, the Israel Ministry of Health recommended supplementing this test with a WES analysis, especially for children with severe ASD and/or those from multiplex ASD families [25]. While genetic testing for ASD has obvious clinical benefits (e.g., facilitating early intervention and, hence, improved outcomes), its economic implications are currently not clear.

Raising a child with ASD involves a substantial financial burden on the family, further exacerbating the physical, psychological and social difficulties associated with ASD [26,27,28]; for example, in an Australian study published in 2014, the median cost to the family of a child with ASD was estimated to be USD 23,700 per year, 90% of which was due to a loss of employment hours [29]. In another study that was performed in the United States and the United Kingdom, the lifetime cost of supporting an individual with ASD exceeded USD 2.2 million [30]. In Israel, the child allowance (paid by the National Insurance Institute of Israel) for a child with ASD can reach USD 15,600 per year, constituting 188% of the regular child allowance [31]_._ This payment is made to help the child’s family partially cover the costs associated with raising a child with ASD. At the age of 18 years, the allowance paid to the person with ASD may be significantly increased to USD 2760 per month [32,33] to cover the daily expenditure of a disabled person, if it is proven that the disabled person has limited earning possibilities.

In health systems management, a cost-effectiveness analysis is an economic assessment tool that is used to quantify the gains in population health due to a specific intervention or policy implementation [34]. In this analysis, the gains are typically assessed by disability-adjusted life year (DALY) or quality-adjusted life year (QALY), indices that incorporate morbidity and mortality, and, therefore, cover both quality of life and life span. Costs include both direct and indirect costs, and comparisons can be made against alternative strategies or against no intervention at all. At present, there is not sufficient research regarding the cost effectiveness of genetic testing of ASD (e.g., [35]), primarily because data about the yield of next-generation sequencing approaches (e.g., WES) in detecting ASD susceptibility variants are sparse, and there is also a lack of data about the lifelong outcomes of a genetic-based ASD diagnosis. In this study, we applied WES analysis to a population-based sample of children with ASD from southern Israel to assess the yield of WES in detecting genetic variants associated with ASD, and to estimate the cost effectiveness of WES findings for families with children with ASD.

## 2. Materials and Methods

### 2.1. Study Sample and Statistical Analysis

The study sample was drawn from children who were diagnosed with ASD at the National Autism Research Center of Israel (NARCI) [36,37], which is located in Be’er Sheva, the capital city of the southern region, between January 2015 and December 2019. The sample included all the children whose parents consented to provide DNA samples for themselves and the children. Based on our clinical records, none of the parents in the study have been diagnosed with ASD, intellectual disability, or any other type of neurodevelopmental disorder. The study sample thus comprised 182 children with ASD (135 Jewish and 47 Bedouin) from 169 families. Of these, one Bedouin family had three affected children, 11 families (10 Jewish and 1 Bedouin) had two affected children, and the remaining 157 families each had one affected child. Demographic and clinical variables were compared between children included in this study and other children in the NARCI database, as well as between children with and without identified genetic variants by WES using standard univariate statistics. Specifically, Mann–Whitney rank sum test was used to assess group differences in continuous and ordinal variables while chi-square or Fisher exact tests were used to assess differences in nominal variables. All statistical tests were performed in IBM SPSS statistical software version 23.

### 2.2. Whole-Exome Sequencing

Genomic DNA was purified from saliva samples collected from the affected children and their parents with Genotek OG-500 and OG-575 saliva collection kits. The DNA samples were sent for WES at the Broad Institute (USA) as part of the autism sequencing consortium (ASC) initiative, as described before [38]. WES results for each DNA sample were obtained in both bam file format (*.bam) and in variant call file format (*.vcf), with the latter format summarizing the variant calls and annotation of the entire sample.

### 2.3. Identification of Candidate ASD Genetic Variants

We developed a bioinformatics pipeline to analyze the WES data with the goal to identify ASD susceptibility variants in genes robustly associated with ASD according to the SFARI gene database [39] (all genes with score 1 as of January 2021 (*n* = 205) listed in Appendix A). We restricted our search to this high-confidence ASD gene panel to assess the current lowest bounds of WES yield and its cost-effectiveness for ASD diagnosis. The pipeline was applied to the multi-sample VCF file that was generated according to the GATK (Genome Analysis Toolkit) best practice guidelines [40]. First, we removed variants with a minor allele frequency of >5% and also variants with low sequence coverage (DP ≤ 20) or low genotype quality (GQ ≤ 50). Then, we used the Ensembl Variant Effect Predictor (VEP) [41] to assess the effect of the variant on the protein sequence. We only considered variants with the following protein sequence effects: stop gained/lost, start lost, splice acceptor/donor, frameshift, and nonsynonymous missense variants with a “deleterious effect” according to either SIFT [42] or PolyPhen-2 [43]. Next, we evaluated the segregation of each of the remaining variants in each family trio (proband and parents) to identify only those variants with proband-specific genotypes (i.e., de novo, recessive, and compound heterozygote, and for male probands X-linked). Of note, de novo variants that appeared in more than two individuals were removed from the analysis, as these are likely to be false positives. Finally, we visually validated (manual curation) the existence of the remaining variants using IGV software [44].

### 2.4. Cost-Effectiveness Analysis

We calculated the incremental cost-effectiveness ratios (ICERs) [45] of a genetic diagnosis for ASD for the following three different diagnostic strategies: (1) CMA alone; (2) WES alone; (3) CMA followed by WES for those with inconclusive CMA findings (CMA + WES). ICER is the ratio between the difference in cost and the difference in outcomes between two interventions (i.e., ICER = ∆Costs/∆Outcomes). The cost of the genetic diagnosis in each scenario included the cost of the genetic test (comprising laboratory expenses and bioinformatics costs) [46,47], the cost of genetic counseling [48], and the parents’ productivity cost (total time spent in clinical consultations) based on the average salary/hour in Israel on 1 January 2021 [49] and assuming that both parents were employed and had to spend two work hours for the genetic test. The details of these costs are presented in Appendix A. We used the genetic yield of each genetic strategy as the outcome measure for its ICER calculation. Specifically, we used the reported CMA diagnostic yield of 10% [50], the WES diagnosis yield found in this study, and the WES + CMA yield, assuming no overlap between WES and CMA findings. For each of the three genetic diagnosis strategies, ICER was calculated using “ASD diagnosis without genetic test” as the reference. Thus, the differences in ICER values between the genetic tests represent the actual difference in their cost effectiveness.

## 3. Results

The WES analysis was completed for 182 of the 872 children with ASD (20.9%) who were registered in the NARCI database at the time of the study (February 2021). Table 1 presents the basic demographic and clinical characteristics of this sample of 182 children in comparison with the other children in the NARCI database. Children with exome data vs. children without exome data were diagnosed earlier (3.01 ± 1.43 vs. 3.35 ± 1.37 years, respectively; *p* = 0.0004), had a higher Autism Diagnostic Observation Schedule (ADOS™) comparison score (7.47 ± 2.3 vs. 6.65 ± 2.3; *p* < 0.0001), and were characterized by more severe ASD symptoms, according to both DSM-5 severity criteria A (49.4% vs. 39.4%; *p* = 0.0014) and B (38.0% vs. 27.0%; *p* < 0.0001). These differences suggest that the parents of children with more severe ASD symptoms are more interested in understanding the genetic causes of ASD in their children.

### 3.1. Genetic Findings

Overall, 32 ASD candidate variants were detected in 28 children, corresponding to an overall diagnostic yield of 15.4% of the WES analysis. In four of these children, two ASD candidate variants were found, each variant in a different gene. Eight of the detected variants (25%) were protein-truncating variants (five frameshift and three nonsense variants), and the remaining (75%) were missense variants. In addition, 10 (26%) de novo variants were detected, 15 variants (47%) showed an autosomal recessive segregation (with four of them in the form of compound heterozygotes), and another seven variants (22%) were found in genes on the X-chromosome and showed X-linked segregation with ASD. Additional details about the 32 identified variants, including, for example, their exact genomic and protein sequence positions, are given in Appendix A.

Table 2 presents the sociodemographic and clinical characteristics of the children with positive and negative findings in the WES analysis. Interestingly, the Bedouin children comprised 46.4% of the children with WES-positive findings, more than twice their portion in the group of children with negative WES findings (22.1%). This means that the WES yield was significantly higher in the Bedouin children than in the Jewish children (27.7% vs. 11.1%, respectively; *p* = 0.036). No other differences were observed between the groups of children with and without WES-positive findings.

### 3.2. Cost-Effectiveness Analysis

The results of the cost effective analysis for the different genetic tests are presented in Table 3. The most cost-effective genetic diagnosis strategy for ASD in Israel today, as evaluated by the ICER, is the CMA (ICER = USD 117 per child), followed by the combined CMA + WES strategy (ICER = USD 124.8 per child), and then by WES alone (ICER = USD 147.4 per child).

We then calculated the prospective ICER for both WES and CMA + WES for different increasing yields and decreasing costs of the WES analysis (Figure 1). As expected, the ICER for both WES alone and the WES + CMA approach decreased with either an increase in the WES yield or a decrease in the WES cost. Consequently, even a slight increase in the WES yield (from 15% to 20%) or a decrease of 20% in the cost of WES (from USD 2000 to USD 1600) would make the combined CMA + WES approach the most cost-effective approach for all, and an additional increase in the yield or decrease in the cost (or both) would make WES alone the most cost-effective approach. Notably, at a WES yield of 25% and a cost of USD 2000, WES alone and CMA + WES would have an identical ICER value of USD 91 per child.

## 4. Discussion

### 4.1. Whole-Exome Sequencing Yield

The ASD WES diagnostic yield of 15.4% that was obtained in this study is consistent with the 8–25% WES yield obtained for children with ASD in other similar studies around the world [34,48]. Obviously, relaxation of the strict criteria used for the identification of candidate ASD variants in this study would have increased the WES yield, but it could have also led to a higher rate of false-positive findings. Unfortunately, today, there are no clear diagnostic criteria for the WES analysis of ASD. The guidelines of the American College of Medical Genetics and Genomics (ACMG) for the interpretation of sequence variants [51] that are usually used for this purpose are largely applicable to inherited Mendelian disorders, and are, therefore, less appropriate for multifactorial conditions, such as ASD [52,53]. Obviously, ongoing and future genetic studies of ASD will lead to the identification of additional genes associated with ASD, and to an improved understanding of the genetic etiology of ASD, which, in turn, will increase the diagnostic yield of WES for ASD.

### 4.2. Ethnic Differences in Whole-Exome Sequencing Yield

The finding of significant differences in the WES yields between the Bedouin and Jewish families suggests that there may be different genetic causes of ASD in these two populations. This finding is not surprising, given the remarkable genetic differences between these two populations [54]. Specifically, the Bedouin, who live mainly in southern Israel and represent about 3.5% of the country’s population [55], constitute a highly inbred community (due to the high rates of consanguineous marriages [56]), resulting in relatively high incidences of specific genetic disorders [56], and possibly also in the aggregation of ASD susceptibility genetic variants in certain families. The higher rate of children with severe ASD in the Bedouin population, as reported previously [57], could partially derive from this lifestyle, although it could also be attributed to an underdiagnosis of milder types of ASD in this population [58].

### 4.3. Cost Effectiveness of Whole-Exome Sequencing for Children with ASD in the Israeli Health System

Our data show that, currently, the most cost-effective strategy for ASD genetic diagnosis in Israel is CMA alone. Nevertheless, our results suggest that adding WES analysis for children with negative findings in the CMA analysis will almost triple the diagnostic yield of the genetic test for ASD, with only a minor increase in the ICER of USD 7.8 per child. Importantly, the ICER difference between CMA alone and CMA + WES will close if there is a reduction in WES costs of a minimum of 25% (to USD 1600) or a 30% increase in the WES yield (to 20%). Importantly, we note that this is, in fact, the current situation in the Bedouin population, for which the WES diagnostic yield is 27.6%. An additional reduction in WES costs or an increase in yield will make WES alone the most cost-effective genetic test for ASD. In such a scenario, we should perhaps consider making WES the first-choice genetic test for children diagnosed with ASD and add a CMA analysis only for children with negative WES findings.

An important aspect that should be considered in the evaluation of the cost effectiveness of WES for children with ASD is the reduction in lifelong financial burden resulting from positive findings in such a genetic test. In this realm, it is reasonable to assume that genetic tests will facilitate earlier and more robust ASD diagnosis, which will, in turn, lead to better intervention outcomes that will improve the child’s chances to join the workforce. As an adult, that child, diagnosed early with ASD, would be able to support him/herself partially or completely [59], thereby saving future costs, both for the family and for the health system. Nevertheless, an assessment of the financial benefit associated with such a genetic diagnosis is extremely complicated and is beyond the scope of this study.

### 4.4. Additional Implications of Whole-Exome Sequencing for Children with ASD

The results of postnatal WES analysis may have additional, non-financial implications for the families of children with ASD. The identification of an ASD susceptibility genetic variant in an affected child may help the parents in their future family planning. This, in fact, has been noted as the main factor for families to conduct such postnatal genetic tests [60]. The WES results may also help to better understand the molecular mechanism underlying ASD in affected children and consequently help guide precision medicine for these children [61]. Nevertheless, using WES as a diagnostic tool for ASD may also have some limitations that should be considered. The major limitation of this approach is its current low diagnostic yield. This means that many families will receive null results that might be false negative. Furthermore, despite the use of stringent criteria for the identification of ASD susceptibility variants in WES analysis, a positive WES finding could still be a false positive. Thus, performing an additional validation of WES-positive findings is highly recommended. Altogether, all these financial and non-financial implications should be considered and better communicated to families of children with ASD.

## 5. Conclusions

Our findings suggest that WES is a relevant and important method for identifying the possible genetic cause in >15% of children with ASD, and its diagnostic yield is expected to increase in the close future as more genes associated with ASD are identified. The indispensable clinical and financial benefits of such genetic findings, together with the predicted reduction in the ICER of this test in the next few years, would support a recommendation for wider utilization of this approach for clinical purposes in Israel and elsewhere.

## Figures and Tables

**Figure 1 genes-13-00036-f001:**
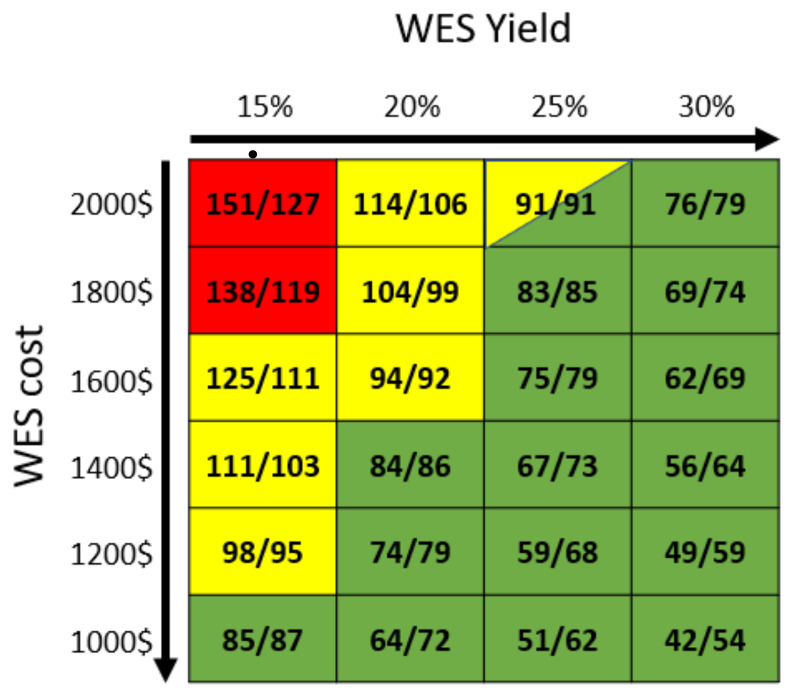
Incremental cost-effectiveness ratio (ICER) of WES alone and of CMA + WES strategies for different WES yields and costs. The ICERs of the WES strategy (numbers on the left) and of the CMA + WES (numbers on the right) strategy are depicted for different WES yields (X-axis) and costs (Y-axis). Cell colors indicate the most cost-effective strategy for each WES yield–cost combination (red, CMA alone; yellow, CMA + WES; green, WES alone).

**Table 1 genes-13-00036-t001:** Characteristics of the 872 children with ASD in the NARCI database.

Variable		Children with Exome (*N* = 182)	Children without Exome (*N* = 690)	*p*-Value
Gender (Male)		139 (76.4%)	555 (80.4%)	0.2844 ^a^
Ethnicity (Bedouin)		47 (25.8%)	165 (23.9%)	0.5512 ^a^
Diagnosis age (years; mean, SD)		3.01 (1.43)	3.35 (1.37)	0.0004 ^b^
IQ (mean, SD)		72.71 (18.79)	75.37 (17.89)	0.1516 ^b^
ADOS module (*N* = 705)	Toddler	57 (39.3%)	150 (26.8%)	0.0165 ^c^
	1	57 (39.3%)	239 (42.7%)	
	2	18 (12.4%)	111 (19.8%)	
	3	13 (9.0%)	60 (10.7%)	
ADOS comparison score (mean, SD)		7.47 (2.3)	6.65 (2.3)	<0.0001 ^b^
DSM-5 severity level (A) ^#^ (*N* = 782)	1	12 (7.6%)	112 (17.9%)	0.0014 ^c^
	2	68 (43.0%)	266 (42.6%)	
	3	78 (49.4%)	246 (39.4%)	
DSM-5 severity level (B) ^#^ (*N* = 782)	1	15 (9.5%)	145 (23.2%)	<0.0001 ^c^
	2	83 (52.5%)	310 (49.7%)	
	3	60 (38.0%)	169 (27.0%)	

Values are numbers of participants, with percentages in parentheses, unless specified otherwise. ^#^ DSM-5 severity levels: 1—“Requiring support”; 2—“Requiring substantial support”; 3—“Requiring very substantial support”. ^a^ χ^2^ test; ^b^ Mann-–Whitney U test; ^c^ χ^2^ linear-by-linear test.

**Table 2 genes-13-00036-t002:** Characteristics of 182 children with ASD for whom WES results were available.

Variable		Children with Positive WES Findings (*N* = 28)	Children with Negative WES Findings (*N* = 154)	*p*-Value
Gender (Male)		20 (71.4%)	119 (77.3%)	0.503 ^a^
Ethnicity (Bedouin)		13 (46.4%)	34 (22.1%)	**0.036 ^a^**
Diagnosis age (years; mean, SD)		2.62, 0.90	3.08, 1.50	0.114 ^b^
IQ (mean, SD)		69.6, 18.42	73.2, 18.89	0.38 ^b^
ADOS module (*N* = 145)	Toddler	12 (50%)	45 (37.2%)	0.162 ^c^
	1	9 (37.5%)	48 (39.7%)	
	2	2 (8.3%)	16 (13.2%)	
	3	1 (4.2%)	12 (9.9%)	
ADOS comparison score (mean, SD)		8.04, 2.44	7.36, 2.252	0.073 ^b^
DSM-5 severity level (A) ^#^ (*N* = 158)	1	3 (11.5%)	9 (6.8%)	0.33 ^c^
	2	12 (46.2%)	56 (42.4%)	
	3	11 (42.3%)	67 (50.8%)	
DSM-5 severity level (B) ^#^ (*N* = 158)	1	2 (7.7%)	13 (9.8%)	0.412 ^c^
	2	17 (65.4%)	66 (50%)	
	3	7 (26.9%)	53 (40.2%)	

Values are numbers of participants, with percentages in parentheses, unless otherwise specified. ^#^ DSM-5 severity levels: 1—“Requiring support”; 2—“Requiring substantial support”; 3—“Requiring very substantial support”. ^a^ χ^2^ test; ^b^ Mann–Whitney U test; ^c^ χ^2^ linear-by-linear test. Bold font represents significant differences.

**Table 3 genes-13-00036-t003:** Cost-effective analysis of different genetic diagnosis strategies.

Strategy	Total Cost (USD)	Outcome–Diagnostic Yield by Each Strategy (%)	Incremental Cost-Effectiveness Ratio (ICER)
CMA	1170	10	117.0
WES	2270	15.4	147.4
CMA + WES	3170	25.4	124.8

## Data Availability

WES data were generated as part of the ASC and are available in dbGaP with study accession: phs000298.v4.p3.

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
