# Peer review of "Diagnostic Yield and Economic Implications of Whole-Exome Sequencing for ASD Diagnosis in Israel"

_genes, 2021, doi:10.3390/genes13010036_

Round 1

Reviewer 1 Report

I felt this was a very interesting paper that discussed the cost effectiveness of using WES in diagnosing ASD. Unfortunately, however, I thought that it would be difficult to argue for similar effects in all countries. There are two points that I hope you will revise.

  1. As you know, ASD is a multifactorial disease, and I believe that there are significant hereditary factors from parents to children, as well as environmental factors. What information do you have about the parents of this patient (whether they have been diagnosed with ASD or not, whether they have intellectual disabilities or not, etc.)? It seems to me that this information is necessary for interpreting the WES results.
  2. I would like to know more about the pros and cons of diagnosing ASD based on the presence of genetic mutations. It is a multifactorial disease, and as you mentioned in your paper, I feel that it is still a topic under discussion.

Author Response

Dear Reviewer,

We would like to thank you for the time and effort you put in reviewing our manuscript. We carefully considered each of your comments while rewriting our manuscript and believe it has improved significantly. Please find below your comments, each follows by our response. We hope that the revisions in the manuscript and our accompanying responses addressed all of your concerns regarding this manuscript and will make it suitable for publication in GENES. We would be glad to respond to any further questions and comments that you may have.

Sincerely,

The authors

Reviewer comments: 

I felt this was a very interesting paper that discussed the cost effectiveness of using WES in diagnosing ASD. Unfortunately, however, I thought that it would be difficult to argue for similar effects in all countries. There are two points that I hope you will revise.

  1. As you know, ASD is a multifactorial disease, and I believe that there are significant hereditary factors from parents to children, as well as environmental factors. What information do you have about the parents of this patient (whether they have been diagnosed with ASD or not, whether they have intellectual disabilities or not, etc.)? It seems to me that this information is necessary for interpreting the WES results.

Author reply: We agree that this is an important information that we missed in the original manuscript. Therefore, we added text to the “study sample” section in the Methods (page 3) stating that “Based on our clinical records, none, of the parents in the study have been diagnosed with ASD, intellectual disability, or any other type of neurodevelopmental disorder.”

  1. I would like to know more about the pros and cons of diagnosing ASD based on the presence of genetic mutations. It is a multifactorial disease, and as you mentioned in your paper, I feel that it is still a topic under discussion.

Author reply: We thank the reviewer for this important comment. We added a whole new paragraph to the discussion (page 9) titled: “Additional implications of Whole-Exome Sequencing for Children with ASD” where we discuss additional pros and cons of using WES to facilitate ASD diagnosis.  

Reviewer 2 Report

In this study the authors did report an interesting and important analysis of cost-effectivness of different genetic approach to  ASD diagnosis. It is clear that precocious genetic analysis represent the most important tool for ASD diagnosis and to perform early treatment that is crucial especially in more serious cases. In particular they assessed coste effectivness on the bassi of cost and yield of CMA, WES and CMA+WES approaches . They concluded that  WES analysis seem to facilitate ASD diagnosis  in Israel, especially for highly consanguineous populations, such as the Bedouin.

This is a very intersting study which may be useful for future strategies in ASD diagnosis. However I have some concerns regarding the statistical analysis. Only a brief citation of chi square test used was reported as legend of tables 1 and 2 , a more detailed decription may be reported in a specific paragraph.

Moreover cost effective analysis has been reported as descriptive report in table 3 and figure 1 but no statistical evalaution were performed. A statistical evaluation of the difference between different approaches would give more objective results and improve the conclusions.

Author Response

Dear Reviewer,

We would like to thank you for the time and effort you put in reviewing our manuscript. We carefully considered each of your comments while rewriting our manuscript and believe it has improved significantly. Please find below your comments, each follows by our response. We hope that the revisions in the manuscript and our accompanying responses addressed all of your concerns regarding this manuscript and will make it suitable for publication in GENES. We would be glad to respond to any further questions and comments that you may have.

Sincerely,

The authors

--------

Reviewer comments: 

In this study the authors did report an interesting and important analysis of cost-effectivness of different genetic approach to  ASD diagnosis. It is clear that precocious genetic analysis represent the most important tool for ASD diagnosis and to perform early treatment that is crucial especially in more serious cases. In particular they assessed coste effectivness on the bassi of cost and yield of CMA, WES and CMA+WES approaches . They concluded that  WES analysis seem to facilitate ASD diagnosis  in Israel, especially for highly consanguineous populations, such as the Bedouin.

  1. This is a very intersting study which may be useful for future strategies in ASD diagnosis. However I have some concerns regarding the statistical analysis. Only a brief citation of chi square test used was reported as legend of tables 1 and 2 , a more detailed description may be reported in a specific paragraph.

Author reply: We added text to the end of the 1st paragraph of the Methods section (page 3) that describes the statistical tests used in the study, and also revised the title of this paragraph to: “Study Sample and statistical analysis”

  1. Moreover, cost effective analysis has been reported as descriptive report in table 3 and figure 1 but no statistical evalaution were performed. A statistical evaluation of the difference between different approaches would give more objective results and improve the conclusions.

Author reply:  Thank you for your comment. We now understand the potential confusion a reader might have regarding these results. The ICER calculation in this manuscript is based on real world data and therefore represents genuine and accurate cost-effectiveness differences. Thus, there is no need in statistical tests to assess the significance of these differences. We added text to the end of the “Cost-effectiveness analysis” in the Methods (page 4) to clarify this issue. Of note, our sensitivity analysis depicted in Figure 1, provides a broader and more objective view of the differences between genetic test strategies assuming potential changes in the costs and yield of WES.

Round 2

Reviewer 2 Report

The manuscript was improved and the  authors assessed all my questions